# Empower-Grief for Relatives of Cancer Patients: Implementation and Findings from an Exploratory Randomized Controlled Trial

**DOI:** 10.3390/bs15070972

**Published:** 2025-07-17

**Authors:** David Dias Neto, Alexandra Coelho, Ana Nunes da Silva, Teresa Garcia Marques, Sara Albuquerque

**Affiliations:** 1APPsyCI—Applied Psychology Research Center Capabilities & Inclusion, ISPA—Instituto Universitário, 1149-041 Lisbon, Portugal; manuela.coelho@ispa.pt; 2Centro de Investigação em Ciência Psicológica (CICPSI), Faculdade de Psicologia, Universidade de Lisboa, 1649-004 Lisbon, Portugal; acsilva@psicologia.ulisboa.pt; 3WJCR—William James Center for Research, ISPA—Instituto Universitário, 1149-041 Lisbon, Portugal; gmarques@ispa.pt; 4HEI-Lab: Digital Human-Environment Interaction Labs, Lusófona University, 1749-024 Lisbon, Portugal; sara.albuquerque@ulusofona.pt

**Keywords:** prolonged grief disorder, palliative care, Empower-Grief, psychological intervention, bereavement

## Abstract

Grief reactions among relatives of palliative care patients are often overlooked, with most interventions targeting Prolonged Grief Disorder (PGD) rather than its prevention. Few interventions have been developed for individuals at risk. This study aimed to evaluate the efficacy of Empower-Grief, a selective intervention designed to address early problematic grief reactions and to explore predictors of its effectiveness. This exploratory randomized controlled trial (RCT) compared Empower-Grief with Treatment as Usual (TAU) among relatives or caregivers of palliative and oncological patients at risk of developing PGD. A total of 46 participants were assessed at baseline, post-intervention, and six months later. The primary outcome was PGD symptoms, with additional measures including anxiety, depression, coping strategies, attachment style, psychological flexibility, post-traumatic growth, social support, and therapeutic alliance. The final analyses indicate equivalence between Empower-Grief and TAU, suggesting that both interventions yielded comparable outcomes in reducing PGD symptoms and associated psychological distress. The initial symptoms and therapeutic alliance were predictors of the results in both post- and follow-up moments. This study contributes to the evidence on grief interventions in palliative care, highlighting the importance of structured support for bereaved caregivers. While Empower-Grief demonstrated comparable effectiveness to TAU, its lower intensity, ease of training, and application make it a promising treatment option.

## 1. Introduction

Grief is a natural and universal experience, but for a sizable subset of bereaved individuals, the mourning process can develop into a clinically significant and prolonged condition. Prolonged Grief Disorder (PGD) has recently gained prominence in psychiatric diagnostics, particularly with its inclusion in the International Classification of Diseases, 11th revision (ICD-11), and in the Diagnostic and Statistical Manual of Mental Disorders, fifth edition, text revision (DSM-5-TR). Recently, the DSM-5-TR extended this period to one year post-loss ([2]).

PGD is increasingly recognized as a significant public health concern due to its prevalence (10–20%) and associated societal burdens ([11]). The impact of PGD extends well beyond individual mental health, affecting social and occupational functioning. Individuals suffering from PGD often experience significant disruptions in their daily lives, including difficulties in engaging in social relationships and returning to work ([31]; [50]). The research indicates that PGD is correlated with decreased quality of life, increased work absenteeism, and a greater likelihood of using healthcare services compared to both not bereaved individuals and those experiencing normal grief reactions ([23]; [26]; [44]). Specifically, PGD has been linked to an elevated risk of mental health disorders, including depression and anxiety, which further contribute to the social and economic burden associated with this condition ([23]; [38]).

High levels of PGD (e.g., 28% at 13 months) have been identified among caregivers both worldwide (e.g., [20]) and in Portugal ([10]), highlighting cancer-related bereavement as a unique context of emotional vulnerability. The intricate dynamics surrounding caregiving and the characteristics of cancer loss, particularly its prolonged course and the multifaceted changes in family dynamics, complicate the bereavement experience. For instance, [30] ([30]) found that a lack of choice in caregiving roles significantly predicted greater emotional stress among cancer caregivers. Also, the unpredictability of cancer progression was found to contribute significantly to caregiver anxiety, with the compounding effects of dual caregiving roles and shifting family dynamics exacerbating emotional strain ([52]). Consequently, understanding the multifaceted emotional landscape of cancer-related bereavement and caregiver experiences is crucial for developing supportive interventions aimed at alleviating the burdens associated with this unique context. A range of psychological interventions has been developed to support individuals experiencing grief, including supportive counseling, cognitive-behavioral therapy (CBT), acceptance and commitment therapy (ACT), and mindfulness-based approaches ([13]; [19]; [29]). Meta-analyses have demonstrated their effectiveness in alleviating symptoms of grief, depression, and anxiety, particularly when interventions are delivered by trained professionals over multiple sessions ([21]; [25]). However, these interventions tend to focus on individuals already experiencing severe or prolonged grief, leaving a critical gap in early, preventive support for those at moderate risk.

According to the public health model of bereavement support ([3]), most bereaved individuals fall within a moderate-risk category, experiencing distress that may not meet diagnostic thresholds but still poses a threat to psychological well-being. This group frequently lacks access to structured, evidence-based interventions, particularly within routine healthcare systems that prioritize either informal support or clinical treatment for high-risk cases ([41]). Additionally, bereavement care in real-world hospital settings has multiple challenges, both at the organizational and practice levels. Hospital-based bereavement support is often hindered by logistical, infrastructural, and human resource constraints that impede the consistent delivery of high-quality care. In hospital-based settings, bereavement care is often undermined by high workloads, space limitations, and a lack of trained personnel.

Treatment as usual (TAU) practices, that is, default or routine care provided without the benefit of specialized interventions, are the general norm. In many hospital settings, TAU is marked by pronounced variability, partly due to the difficulties of compliance with standardized care guidelines (e.g., Norma nº 003/2019 de 23/04/2019—Direção-Geral da Saúde) and the absence of robust evaluation frameworks, which are challenges that has been noted worldwide ([1]; [3]). Consequently, bereaved individuals may receive minimal or uneven support, and the gap between recommended practices and actual delivery persists. Moreover, while high-risk groups occasionally receive targeted interventions, a significant proportion of moderately at-risk individuals are left with generic and less effective TAU protocols that do not adequately address their preventive care needs ([1]; [3]). Additionally, caregivers frequently report feeling overlooked and inadequately supported during and after the end-of-life phase, with many stating that their emotional needs are not systematically addressed due to the absence of dedicated post-death follow-up services ([4]). Such inconsistent, unstandardized, and reactive (rather than preventive) practices highlight an urgent need for the development of tailored interventions, standardized care pathways, and consistent evaluation measures to bridge the gap between policy guidelines and day-to-day practice ([1]; [46]).

The EMPOWER intervention program (Enhancing and Mobilizing the POtential for Wellness and Emotional Resilience) was initially developed to address the urgent mental health needs of surrogate decision makers in ICUs—individuals who are tasked with making life-and-death decisions on behalf of incapacitated patients in intensely stressful environments. These surrogates are at high risk for PTSD, depression, PGD, and decisional regret ([45]; [28], [27]).

At its theoretical core, EMPOWER integrates cognitive-behavioral and acceptance-based models of grief. Cognitive-behavioral approaches target maladaptive thought patterns and promote cognitive restructuring, while acceptance-based strategies—including mindfulness practices—focus on enhancing psychological flexibility and nonjudgmental awareness of grief-related emotions. A specific target of EMPOWER is experiential avoidance, which refers to the tendency to suppress or avoid painful emotions and thoughts. Experiential avoidance has been empirically linked to high anticipatory grief ([15]; [42]) and a range of psychological disorders, including PGD, depression, and anxiety (e.g., [16]; [49]; [54]). Evidence suggests that experiential avoidance moderates the association between motivational sensitivity and prolonged grief symptoms, indicating that individuals who persistently avoid painful internal experiences are more susceptible to intensified grief reactions. This finding directly links experiential avoidance to the severity of PGD symptoms, highlighting its role as a maladaptive emotion regulation strategy that can impede the processing of loss.

Reducing experiential avoidance is viewed in this program as a central mechanism of change, enabling individuals to engage more openly with their emotional pain and foster acceptance and resilience ([28]; [22]). This program consists of six brief, modular sessions (totaling 1.5–2 h), followed by two booster sessions, and has demonstrated strong feasibility, high acceptability, and promising reductions in distress, including PGD and experiential avoidance ([28], [27]).

Building on this foundation, Empower-Grief was developed as an early preventive intervention specifically for moderate-risk bereaved caregivers within 3 to 12 months post-loss ([39]). Adapted from the original EMPOWER framework, Empower-Grief retains its core structure and theoretical foundation but is tailored to a new population and context: bereavement in oncology and palliative care. Delivered across six weekly sessions, with two optional booster calls, this intervention combines psychoeducation, cognitive restructuring, mindfulness, and experiential techniques, such as memory-making and imagined dialogue ([53]; [32]). The intervention is manualized, culturally adapted, and delivered by trained professionals under clinical supervision, with strict fidelity monitoring ([39]).

The current exploratory randomized control study has two primary objectives. First, it aims to evaluate the preliminary effectiveness of Empower-Grief compared to treatment as usual (TAU) in reducing PGD symptoms and psychological distress (i.e., anxiety and depression) in family caregivers of palliative and oncological patients. We expected Empower-Grief patients to show reductions in grief symptoms, anxiety, and depression over time and that these changes would be comparable to or greater than those observed in the TAU group. Second, it seeks to explore predictors of intervention response, drawing from the established literature. The considered variables are initial symptom, social support, attachment style, coping, psychological flexibility, and the therapeutic alliance.

By addressing grief early and proactively, Empower-Grief provides a scalable, evidence-based solution for bereavement support in under-resourced clinical environments. It offers a crucial advancement in developing tailored, equitable, and theory-driven interventions for caregivers at moderate risk of psychological complications following a loved one’s death—especially within palliative care, where evidence-based approaches remain limited and are urgently needed to address the complex emotional needs of the bereaved ([8]).

## 2. Materials and Methods

### 2.1. Participants

Participants were the primary caregivers of deceased patients who had received palliative or oncological care at the Centro Hospitalar Universitário Lisboa Norte (CHULN) in Lisbon, Portugal. They were identified through the hospital’s palliative care and oncology service registries. Two exclusion criteria were established: individuals reporting a diagnosis of severe or active mental disorders, such as schizophrenia, bipolar disorder, and major depression, predating the loss, and those receiving current psychological intervention. The inclusion criteria were adults who had a deceased relative within the last 3 to 12 months and were at moderate to high risk of developing PGD, defined as a score above seven on the Risk Assessment for Grief (RAG) scale ([35]). This approach aimed to provide psychological care only to participants demonstrating relevant distress.

Of the 423 contacts available in the registry, 294 were reachable and accepted the initial triage, 102 were eligible participants, and 46 agreed to participate. Among these, 40 (87.0%) were female, with an average age of 47.7 years (*SD* = 15.18). The education level was high by national standards, with 15 (32.2%) holding a BA or MA and 12 (26.1%) having completed secondary school. Regarding death characteristics, the deceased were mostly the father (13, 28.3%), the mother (17, 37.0%), or the spouse (13, 28.3%), and the death occurred 7.7 (*SD* = 2.88) months ago.

### 2.2. Interventions

#### 2.2.1. Treatment as Usual (TAU)

TAU consisted of supportive psychotherapy delivered in an individual, unstructured, and integrative format. It focused on the development of adaptive coping strategies and cognitive restructuring of grief-related experiences ([55]). TAU was delivered over a 12-week period, with session frequency adjusted based on individual participant needs. The intervention was conducted by three licensed psychologists who had received specific training in bereavement support and had at least five years of experience (M = 12.3, *SD* = 11.02, range = 5–25). The intervention was unconstrained to reflect the usual practice at the Hospital.

#### 2.2.2. Empower-Grief

Empower-Grief ([9]) is a manualized, cognitive-behavioral, and acceptance-based intervention adapted from EMPOWER ([28]) for post-mortem bereavement contexts. The intervention consists of six structured 50-min sessions, delivered to the client either in person or online, along with two booster sessions conducted at two and four weeks post-treatment to reinforce the intervention’s effects. This intervention was conducted by four licensed psychologists with specific training in this treatment. These psychologists were less experienced, having at least one year of training (M = 3.2, *SD* = 2.50, range = 2–7).

Each session followed a structured framework with defined objectives: 1. Initial assessment and adherence facilitation; 2. Psychoeducation and stabilization strategies; 3. Understanding grief through cognitive-behavioral principles; 4. Promoting experiential acceptance of emotions; 5. Imagined dialogue exercises for emotional processing; 6. Coping skills training and resilience building; 7 and 8 Boosting sessions.

### 2.3. Instruments

#### 2.3.1. Sociodemographic Data

For this research, a brief questionnaire was developed to collect demographic data from the participants, including their gender, age, nationality, and level of education.

#### 2.3.2. Risk Assessment for Grief (RAG; [35])

The RAG was used as a screening tool to classify participants according to their risk of developing PGD. This hetero-assessment scale consists of four items evaluating anger, guilt, interpersonal relationships, and overall coping with grief, rated on a five-point Likert scale. Scores classify individuals into low-risk (<7 points), moderate-risk (7–10 points), or high-risk (≥10 points) categories.

#### 2.3.3. Prolonged Grief Scale—Revised (PG13-R; [44])

The PG13-R was the primary outcome measure. This self-report scale, based on DSM-5-TR criteria for PGD, consists of 13 items rated on a five-point Likert scale. The scale has demonstrated high internal consistency, with Cronbach’s α values of 0.83 (Yale), 0.90 (Utrecht), and 0.93 (Oxford). A PGD diagnosis is assigned when a participant scores above 30 points and meets the temporal (12 months post-loss) and functional impairment criteria.

#### 2.3.4. Hospital Anxiety and Depression Scale (HADS; [56])

The HADPS was used to measure anxiety and depressive symptoms. Originally developed for nonpsychiatric hospital populations, this 14-item instrument includes two subscales (anxiety and depression), with scores ranging from 0 to 3 per item. Higher scores indicate greater symptom severity, with scores above 8 suggesting clinically relevant symptoms. The Portuguese adaptation ([40]) demonstrated good reliability, with Cronbach’s α = 0.76 for anxiety and 0.81 for depression.

#### 2.3.5. Brief COPE ([7])

The Brief COPE is a 28-item measure comprising 14 coping strategies, each represented by two items. These strategies include humor, positive reframing, emotional and social support, acceptance, religious coping, planning, active coping, behavioral disengagement, self-blame, substance use, venting, self-distraction, and denial. Items are rated on a four-point Likert scale (0–3), with mean scores calculated for each coping factor. The Portuguese adaptation ([47]) has demonstrated adequate internal consistency, with Cronbach’s α values generally above 0.70.

#### 2.3.6. Experiences in Close Relationships Scale—Short Form (ECR-RS; [17])

The ECR-RS was used to assess attachment to the deceased. This self-report measure consists of nine items rated on a seven-point Likert scale (1–7), assessing attachment-related anxiety (items 1–6) and avoidance (items 7–9). Higher scores indicate greater attachment insecurity. The Portuguese adaptation of the ECR-RS demonstrated adequate reliability (Cronbach’s α = 0.72 to 0.91) and strong construct validity ([37]).

#### 2.3.7. Acceptance and Action Questionnaire-II (AAQ-II; [5])

The AAQ-II is a seven-item measure of experiential avoidance and psychological inflexibility. Responses are given on a seven-point Likert scale (1–7), with higher scores indicating greater psychological inflexibility. The Portuguese validation ([43]) showed strong psychometric properties, including excellent internal consistency (Cronbach’s α = 0.90) and good construct validity.

#### 2.3.8. Multidimensional Scale of Perceived Social Support (MSPSS; [57])

The MSPSS is a 12-item measure that evaluates three dimensions of perceived support (family, friends, and significant others) using a seven-point Likert scale (1–7). Higher scores indicate a greater perceived level of social support. The Portuguese adaptation ([6]) demonstrated high internal reliability (Cronbach’s α = 0.85 to 0.95).

#### 2.3.9. Working Alliance Inventory (WAI-S; [51])

The WAI-S is a 12-item self-report scale that assesses three dimensions of the therapeutic relationship: bond, tasks, and goals. Items are rated on a seven-point Likert scale (1–7), with higher scores indicating a stronger therapeutic alliance. The Portuguese version of the WAI-S demonstrated high internal consistency (Cronbach’s α = 0.89 for the patient version and 0.85 for the therapist version) ([33]).

### 2.4. Procedures

This study followed an exploratory randomized controlled trial (RCT) design, comparing the Empower-Grief intervention with treatment as usual (TAU). The study was approved by the Ethics Committee of ISPA—Instituto Universitário (I-138-2-24). Following SPIRIT guidelines, the trial was preregistered (ClinicalTrials.gov: NCT06270381), and the protocol was published ([39]). Participants were contacted by phone by a research assistant, following a contact protocol, using the contact information from the registry. During the phone call, the study purpose was explained, and the participants were asked to provide the information required to complete the screening procedure (RAG). All participants provided informed consent before enrollment and were informed about the study’s objectives, procedures, potential risks, and benefits. Confidentiality and data security measures were strictly maintained throughout the study.

Randomization was conducted independently by a research assistant using a computer-generated random allocation sequence. A block randomization method was employed to ensure a balanced distribution between the two intervention groups. Randomization occurred only after eligibility had been confirmed, informed consent was obtained, and the baseline assessment was completed. To minimize potential bias, clinicians delivering the interventions were not involved in the randomization process and remained unaware of the participants’ initial risk and baseline assessments.

The study included three assessment time points to evaluate treatment outcomes. The baseline assessment was conducted prior to the initiation of the intervention. The post-intervention assessment took place immediately after the completion of the Empower-Grief program or 12 weeks (i.e., a similar duration) after the baseline evaluation for TAU. The follow-up assessment took place six months after treatment completion. The outcome measures were applied at all three time points, while the predictors were assessed only at baseline.

Several procedures were implemented to ensure treatment fidelity in the Empower-Grief condition. All psychologists delivering the Empower-Grief intervention received extensive training, which included 20 to 30 h of theoretical instruction and supervised practice. Training sessions involved model interventions conducted by a senior grief psychologist. TAU providers adhered to their usual case discussions and supervision practices at the hospital.

### 2.5. Analysis

All statistical analyses were conducted within a Bayesian framework, using Bayesian repeated measures ANOVAs and Bayesian regression models to evaluate treatment effects and identify predictors of change. The specific Bayesian approach used was chosen not only due to the modest sample size, which limits the reliability of frequentist null hypothesis testing, but also because it allows for a more nuanced assessment of change across three time points, enabling direct quantification of evidence for or against specific effects through Bayes Factors (BF_10_). In addition, due to the use of a Bayesian framework, no power analysis was conducted ([24])[note 1]. For the primary and secondary outcomes (prolonged grief, anxiety, and depression), Bayesian repeated-measure ANOVAs were used to examine the main effects of time (baseline, post-intervention, and follow-up), treatment condition (Empower-Grief vs. TAU), and their interaction. To assess the contribution of individual characteristics, baseline variables (e.g., anxiety and depression) and treatment dosage (i.e., the total number of sessions) were included as covariates in the follow-up models. In all analyses, a BF_10_ > 3 was interpreted as moderate evidence and BF_10_ > 10 as strong evidence in favor of an effect.

Since no strong treatment effects emerged, analyses addressing the second study objective—identifying predictors of change—were conducted across the full sample. Bayesian regression models were employed to evaluate the contribution of baseline psychological variables (e.g., attachment style, coping strategies, psychological flexibility, perceived social support, therapeutic alliance) to both short-term change (post – baseline) and long-term change (follow-up – baseline) in grief symptoms. Predictors were entered individually or in combination (e.g., PG13 scores at multiple time points), and results were reported as model inclusion probabilities, Bayes Factors, and explained variance (R^2^). All analyses were performed using Jamovi 2.3.26 (JSQ module for Bayesian Methods).

## 3. Results

The participants had similar demographics across groups. Regarding gender, a Bayesian contingency table analysis yielded a Bayes factor (BF_10_) of 1.57, indicating weak evidence for an association between gender and the treatment group. The TAU group consisted of 21 females and one male, while the Empower group included 19 females and five males. The TAU group had an average age of 47.96 years (*SD* = 16.10), while the Empower group averaged 47.54 years (*SD* = 14.64), with no meaningful differences found in a Bayesian *t*-test (BF_10_ = 0.294). Similarly, the time since the loss was comparable between groups (TAU: M = 7.09, *SD* = 3.19; Empower-Grief: M = 8.25, *SD* = 2.51), with a BF_10_ = 0.626.

Table 1 presents the results of the initial risk assessment for the outcome variables at baseline and subsequent time points. As expected from randomization, no group showed significant differences in their initial assessments. None of the observed differences in potential predictors yielded a Bayes Factor (BF_10_) above 1, indicating little evidence for group differences at baseline. Importantly, the PG-13 scores were slightly lower than those reported in other clinical samples in the literature ([44]). Table 1 also presents the observations of the outcome variables for post-treatment and follow-up moments.

Regarding potential predictors, none of the observed differences reached a Bayes Factor (BF_10_) greater than 1, indicating little evidence for group differences at baseline assessments. The ECRRS scores were similar between the TAU (*M* = 2.237, *SD* = 1.482) and Empower-Grief groups (*M* = 2.380, *SD* = 1.235; BF_10_ = 0.308). Similarly, no strong differences were observed for problem-focused coping (TAU: *M* = 11.227, *SD* = 5.537; Empower-Grief: *M* = 10.292, *SD* = 4.676; BF_10_ = 0.342), emotion-focused coping (TAU: *M* = 16.136, *SD* = 6.213; Empower-Grief: *M* = 13.625, *SD* = 5.355; BF_10_ = 0.698), or avoidant coping (TAU: *M* = 7.091, *SD* = 3.407; Empower-Grief: *M* = 7.583, *SD* = 3.283; BF_10_ = 0.324). The same pattern was found for MSPSS (TAU: *M* = 5.602, *SD* = 1.307; Empower-Grief: *M* = 5.132, *SD* = 1.303; BF_10_ = 0.534), AAQ (TAU: *M* = 37.273, *SD* = 10.682; Empower-Grief: *M* = 41.333, *SD* = 9.168; BF_10_ = 0.634), WAI (TAU: *M* = 52.722, *SD* = 8.477; Empower-Grief: *M* = 52.000, *SD* = 7.330; BF_10_ = 0.338), and IES (TAU: *M* = 43.167, *SD* = 13.448; Empower-Grief: *M* = 38.625, *SD* = 10.436; BF_10_ = 0.520). These results suggest that there were no meaningful baseline differences between groups across all assessed psychological measures.

No differences in overall dropout were found between groups at the 6-month assessment, as indicated by a Bayesian contingency table analysis (BF_10_ = 0.704). A total of 8 participants dropped out of TAU, and 10 participants dropped out of Empower-Grief. A notable difference was found in the total number of sessions (BF_10_ = 1.132), with the TAU participants receiving more sessions (*M* = 8.73, *SD* = 5.31) than the Empower participants (*M* = 6.33, *SD* = 2.66). However, when considering only the initial three-month period—which corresponds to the duration of the Empower program—the results were reversed. The TAU participants (*M* = 4.55, *SD* = 2.02) had fewer sessions than those in the Empower group (*M* = 6.33, *SD* = 2.66), with moderate evidence supporting a difference (BF_10_ = 3.68). This difference in continuous support throughout time was further reinforced by the continuation of six participants in TAU even after 6 months of follow-up. Overall, the Empower-Grief group had a higher intensity in the first three months, whereas the TAU group had a higher intensity overall.

### 3.1. Effects of Treatments Across Time

A Bayesian repeated measures ANOVA was conducted to assess the effects of time, treatment, and their interaction on grief, anxiety, and depression. Figure 1 shows the mean results for the three outcomes—grief symptoms, anxiety, and depression.

Table 2 presents the Bayesian ANOVA results for the considered effects. For grief symptoms, there was moderate evidence for a main effect of time, suggesting significant changes across assessments, but weak evidence for a main effect of treatment and weak evidence against the interaction between time and treatment. The grief scores decreased over time in both groups: at the baseline, the TAU group results were (*M* = 28.1, *SD* = 6.84) and Empower-Grief group results were (*M* = 28.1, *SD* = 7.17); at the post-intervention period, the TAU group results were (*M* = 24.7, *SD* = 7.74) and Empower-Grief group results were (*M* = 24.4, *SD* = 5.26); and at the 6-month follow-up, the TAU group results were (*M* = 23.1, *SD* = 7.78) and Empower-Grief group results were (*M* = 26.4, *SD* = 8.81).

For anxiety symptoms (HADS-Anxiety scores), there was very strong evidence for a main effect of time but weak evidence for a main effect of treatment and weak evidence against the interaction effect. The anxiety levels decreased over time in both groups: at the baseline, the TAU group results were (*M* = 10.71, *SD* = 3.43) and Empower-Grief group results were (*M* = 9.64, *SD* = 2.21); at the post-intervention period, the TAU group results were (*M* = 7.29, *SD* = 3.54) and Empower-Grief group results were (*M* = 8.64, *SD* = 2.65); and at the 6-month follow-up, the TAU group results were (*M* = 7.00, *SD* = 3.04) and Empower-Grief group results were (*M* = 7.86, *SD* = 3.94). For depression symptoms (HADS-Depression scores), there was very strong evidence for a main effect of time, while the main effect of treatment and the interaction effect showed weak evidence. The depression levels decreased in both groups: at the baseline, the TAU group results were (*M* = 8.80, *SD* = 4.93) and Empower-Grief group results were (*M* = 6.71, *SD* = 3.67); at the post-intervention period, the TAU group results were (*M* = 6.13, *SD* = 4.66) and Empower-Grief group results were (*M* = 5.93, *SD* = 2.40); and at the 6-month follow-up, the TAU group results were (*M* = 5.33, *SD* = 4.67) and Empower-Grief group results were (*M* = 5.79, *SD* = 2.64). These findings suggest that changes in grief, anxiety, and depression were primarily driven by time rather than treatment, with no strong evidence supporting differences between the TAU and Empower-Grief groups in their effectiveness.

To evaluate the second hypothesis, baseline severity and treatment dosage (i.e., the number of sessions) were included as covariates in the Bayesian repeated-measures model. The value of these covariates was assessed by comparing models with and without their inclusion. When adjusting for baseline severity, measured by anxiety and depression, the evidence for a main effect of treatment remained weak (BF_10_ = 0.697), being only slightly higher than in the unadjusted model (BF_10_ = 0.436). A similar pattern emerged when the total number of sessions was included as a covariate (BF_10_ = 0.697). Notably, the evidence for a Time × Treatment interaction increased modestly when initial severity (BF_10_ = 3.350) and number of sessions (BF_10_ = 2.019) were included compared to the original interaction model (BF_10_ = 1.720). This suggests that individual differences in baseline symptoms and treatment intensity may partially account for small differences between treatment groups over time. Given the overall weak to moderate evidence and limited effect sizes, subsequent analyses were conducted with the two treatment groups combined.

### 3.2. Predictors of Change in Both Treatments

Two Bayesian regression models were run to examine the predictive value of baseline psychological variables for both short-term (post–baseline) and long-term (follow-up to baseline) changes in grief symptoms. Each model included one or two predictors to isolate their individual contribution (see Table 3). For the short-term change, only the PG13 baseline grief result showed moderate evidence for inclusion, explaining 17.3% of the variance. Other variables, such as emotion-focused coping (*R*^2^ = 9.8%) and working alliance (*R*^2^ = 14.2%), showed anecdotal to moderate evidence, though they did not cross the common threshold (BF_10_ > 3). For long-term change, the combination of the PG13 scores at the baseline and post-treatment periods yielded strong evidence and explained 37.4% of the variance in grief reduction. Other predictors showing some evidence of association with follow-up change included avoidant attachment (*R*^2^ = 16.6%) and working alliance (WAI) (*R*^2^ = 14.8%). All of the remaining predictors showed weak or anecdotal evidence for inclusion (BF_10_ < 1).

## 4. Discussion

Grief interventions typically focus on treating Prolonged Grief Disorder (PGD) after it develops rather than preventing it during the vulnerable early bereavement period. There is a lack of low-intensity, structured interventions tailored to at-risk caregivers in palliative care settings. This study aimed to evaluate the effectiveness of Empower-Grief, a structured, low-intensity psychological intervention, in reducing symptoms of Prolonged Grief Disorder (PGD) among caregivers of deceased cancer patients. It compared Empower-Grief to treatment as usual (TAU) through an exploratory randomized controlled trial and examined predictors of intervention success, with the goal of enhancing early preventive care for bereaved caregivers.

The present study had two goals. Regarding the first, Empower-Grief was as effective as TAU in reducing symptoms of prolonged grief, anxiety, and depression over time. Both interventions showed improvement, but no significant differences were found between them. Additionally, there were no differences in the drop-out rates. These similarities may suggest common factors (e.g., empathy, support) that are relevant in addressing initial adverse grief reactions or specific mechanisms (e.g., trauma elaboration, coping strategies development) that may be shared between the two interventions. The role of common factors is well established for psychological intervention in general and bereavement-related interventions (e.g., [48]). This issue can be explored in future research by assessing these dimensions or examining different components. While the intervention did not show superior clinical outcomes compared to TAU, Empower-Grief offers several practical advantages: it is structured, manualized, brief, and can be delivered by less experienced clinicians with adequate supervision. These features may enhance its scalability and cost-effectiveness, particularly in settings with limited specialized resources.

The identification of predictors of intervention response (the second goal) was conducted for both interventions. The initial severity of grief symptoms was the strongest predictor of short- and long-term improvement. The importance of initial severity aligns with the existing literature ([34]). Some psychological variables (e.g., experiential avoidance, attachment avoidance, and therapeutic alliance) demonstrated moderate associations with outcomes; however, no consistent, strong predictors emerged. We can interpret these results in two ways. Firstly, although they did not reach strong predictive thresholds, their moderate association with outcomes indicates potential mediating roles that warrant further investigation in future mechanistic trials. It is important to consider whether these potential variables are merely correlates, have causal relationships, and if they can serve as personalizing parameters ([12]). Secondly, other trait variables, such as personality (e.g., [18]), or process-like variables, such as rituals (e.g., [36]), can be considered in building a richer predictive model. Additional research is needed to utilize these dimensions in personalizing grief interventions.

The overall equivalence of the results occurred despite slight differences in implementation. Firstly, in terms of intensity, Empower participants attended more sessions during the initial three months. In contrast, TAU participants had a higher total number of sessions across the entire follow-up period. Secondly, the therapists implementing TAU had significantly more experience, both in general and palliative care, than those implementing Empower-Grief. Finally, TAU, as an unstructured intervention, relied more on clinical judgment. These three factors suggest that, despite the equivalence in effectiveness, Empower-Grief can be considered as a promising intervention due to its time-limited and structured approach that can be implemented by therapists with less training. Future studies could also explore further venues for increasing the effectiveness, such as adapting Empower-Grief to a group format.

### Limitations

The study has several limitations. Firstly, it is based on a small sample, which is particularly relevant given the low symptom level presented. Subtle differences between treatments or the effects of the predictors could have been explored further with a larger sample. Secondly, the sample was drawn from a single hospital in Portugal and consisted mainly of female caregivers, which limits its applicability to broader populations. Finally, reliance on self-report measures could have been influenced by biases such as social desirability. These limitations result from the practice-based nature of the data gathering and the exploratory nature of this trial. Lastly, if Empower-Grief is conceived as a preventive intervention, comparisons with no treatment could examine the extent to which Empower-Grief reduces the incidence of PGD cases.

## 5. Conclusions

Overall, Empower-Grief appears to be a promising intervention due to its low intensity and broad applicability. Furthermore, it is particularly suited for early practitioners due to its ease of training and the use of an intervention manual. This is a relevant gain considering the role of practitioners in providing support, an element crucially valued by patients (e.g., [14]). Another essential aspect is its pragmatic value. In the context of limited resources and a scarcity of structured responses for bereaved caregivers, the availability of evidence-based approaches is vital. Considering that Empower-Grief is brief and manualized, it is a viable frontline option for palliative care settings, potentially enhancing service efficiency while delivering outcomes comparable to those of standard care. Future research is needed to establish its effectiveness through larger trials to confirm its potential as a scalable, evidence-based response to a widespread and under-addressed need.

## Figures and Tables

**Figure 1 behavsci-15-00972-f001:**
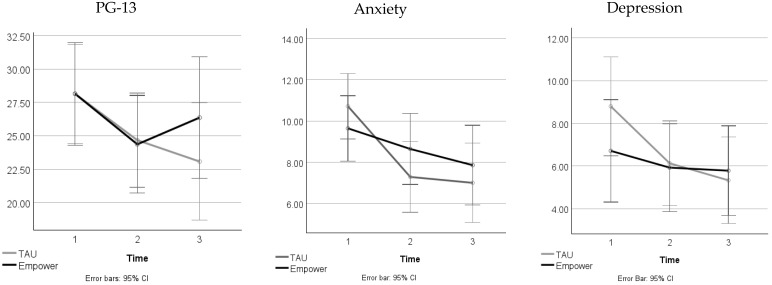
Grief symptoms, anxiety, and depression for TAU and Empower-Grief.

**Table 1 behavsci-15-00972-t001:** Treatment groups comparison of the clinical measures.

	Group	*N*	*M*	*SD*	BF_10_
Risco_Total	TAU	22	8.636	1.293	0.674
	Empower	24	9.250	1.567	
Grief (PG13) 0M	TAU	22	27.955	7.074	0.307
	Empower	24	28.667	6.869	
Grief (PG13) 3M	TAU	18	25.278	8.449	0.356
	Empower	16	24.188	5.128	
Grief (PG13) 6M	TAU	15	23.067	7.778	0.535
	Empower	14	26.357	8.811	
Anxiety (HADS) 0M	TAU	22	11.091	3.531	0.301
	Empower	24	10.833	3.017	
Anxiety (HADS) 3M	TAU	17	7.882	3.839	0.404
	Empower	16	8.688	2.469	
Anxiety (HADS) 6M	TAU	15	7.333	3.200	0.370
	Empower	14	7.857	3.939	
Depression (HADS) 0M	TAU	22	8.636	4.655	0.323
	Empower	24	9.375	5.396	
Depression (HADS) 3M	TAU	18	6.389	4.840	0.353
	Empower	16	5.813	2.509	
Depression (HADS) 6M	TAU	15	5.333	4.670	0.362
	Empower	14	5.786	2.636	

**Table 2 behavsci-15-00972-t002:** Bayesian repeated-measures ANOVA results.

Outcome	Main Effect (Time)	Main Effect (Treatment)	Interaction (Time × Treatment)
Grief (PG-13)	BF_Inclusion_ = 6.729	BF_Inclusion_ = 0.426	BF_Inclusion_ = 0.484
Anxiety (HADS)	BF_Inclusion_ = 340.594	BF_Inclusion_ = 0.509	BF_Inclusion_ = 0.999
Depression (HADS)	BF_Inclusion_ = 92.519	BF_Inclusion_ = 0.827	BF_Inclusion_ = 1.929

**Table 3 behavsci-15-00972-t003:** Predictors of grief change at post-treatment and follow-up.

	Post–Pre BF_10_	*R* ^2^	Follow-up–Pre BF_10_	*R* ^2^
Time since loss	0.352	0.00557	0.509	0.0360
Anxiety (ECR-RS)	0.614	0.0485	0.472	0.0290
Avoidance (ECR-RS)	0.544	0.0393	2.374	0.1656
Total (ECR-RS)	0.284	0.0536	0.984	0.1683
Anxiety (HADS) 0M	0.337	0.00203	0.365	0.00463
Depression (HADS) 0M	0.336	0.00184	0.353	0.00140
Total (HADS) 0M	0.159	0.00490	0.176	0.00544
Emotional coping	1.223	0.0982	0.456	0.02572
Problem-focused coping	0.631	0.0506	0.349	<0.0001
Avoidant coping	0.332	<0.0001	0.355	0.00182
Therapeutic Alliance (WAI)	2.345	0.1420	1.879	0.1475
Psycholog. Flexibility (AAQ)	0.478	0.0295	0.614	0.0533
Social support (MSPSS)	0.432	0.0217	0.553	0.0437
Impact of the event (IES)	0.329	<0.0001	0.362	0.00379
Grief (PG13) 0M	3.83	0.173	0.912	0.0882
Grief (PG13) 3M	-	-	0.693	0.0641
Grief (PG13) 0M + 3M	-	-	18.583	0.3737

## Data Availability

The data are available upon request to the corresponding author.

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
