# Peer review of "Empower-Grief for Relatives of Cancer Patients: Implementation and Findings from an Exploratory Randomized Controlled Trial"

_behavsci, 2025, doi:10.3390/bs15070972_

Round 1

Reviewer 1 Report

Comments and Suggestions for Authors

Thank you for the opportunity to review this manuscript. The study addresses a highly sensitive area—supporting relatives of cancer patients through grief interventions. Specifically, it presents the implementation and preliminary findings of an exploratory randomized controlled trial evaluating the Empower-Grief program. The topic is both relevant and of potential value to clinicians, researchers, and bereavement care providers.

Below are my comments and suggestions aimed at strengthening the manuscript:

  1. Throughout the manuscript, the term Empower and its related explanations are repeated several times. I recommend reducing redundancy by avoiding repetitive descriptions and definitions. Consider synthesizing and streamlining mentions of Empower for clarity and conciseness.
  2. Section 2.3 – Instruments: This section is well-constructed; however, I suggest formatting the content using bullet points or subheadings for each instrument. This would improve readability and allow readers to quickly grasp the structure and content of the measures used.
  3. Please specify the normality test used to verify the distribution of your data (e.g., Shapiro-Wilk or Kolmogorov-Smirnov). This information is necessary to support the appropriateness of parametric analyses.
  4. Include details of the G*Power analysis performed to justify the use of ANOVA and confirm that the sample size was adequate for detecting effects with sufficient statistical power.
  5. I recommend checking and clearly reporting the confidence intervals and associated p-values to strengthen the statistical interpretation and ensure transparency in result reporting.
  6. In the discussion section, it would be beneficial to add a dedicated sub-section discussing the implications of the findings and future directions, emphasizing the study’s significance and its potential contributions to both research and clinical practice.
  7. In addition, I encourage the authors to include a brief discussion of potentially relevant but not directly analyzed variables, such as personality traits, personal religious beliefs, and the presence of rituals that may assist individuals in processing grief. These factors are often reported in the literature as playing a meaningful role in grief trajectories and coping mechanisms. Here is a list of potential references that may help you (there’s a lot of literature about it, so these references may be a starting point):
    • Gana, K., & K'Delant, P. (2011). The effects of temperament, character, and defense mechanisms on grief severity among the elderly. Journal of affective disorders128(1-2), 128-134. (https://pubmed.ncbi.nlm.nih.gov/20619464/)
    • Mitima-Verloop, H. B., Mooren, T. T., & Boelen, P. A. (2021). Facilitating grief: An exploration of the function of funerals and rituals in relation to grief reactions. Death Studies45(9), 735-745. (https://pubmed.ncbi.nlm.nih.gov/31710282/)
  8. It may also be valuable to consider the role of healthcare professionals in providing emotional support. Research suggests that human accompaniment and compassionate care since the diagnosis can significantly support grieving individuals. For instance, even in the devastating context of child loss, many parents do not develop prolonged or complicated grief, especially when surrounded by a supportive and empathetic environment. To further enrich this discussion, you may find it useful to read and cite the following studies and the literature within:
    • Dahò M. (2024) Conscious Decisions in Perinatal Hospice: A Support for Parenting, Family Psychophysical Well-being, and Healthy Grief Processing, Italian Journal of Psychology, 3 (which also includes further references on the role of rituals and personal beliefs). https://www.rivisteweb.it/doi/10.1421/114423
    • Dahò, M. (2020). ‘It was a blanket of love’: How American and Italian parents represent their experience of perinatal hospice through the use of metaphors. Bereavement Care39(3), 112–118. https://www.tandfonline.com/doi/full/10.1080/02682621.2020.1828720 (which explores how grieving parents describe their experiences within a hospice, highlighting the importance of compassionate care by all health workers).

These additions could offer important perspectives that broaden the interpretive depth of the study’s findings. 

  1. Consider adding a dedicated "Limitations" section to discuss methodological or contextual constraints and offer guidance for future research.
  2. Include a Conclusion section that clearly summarizes the main findings, their implications, and potential directions for future investigations. (Last part of the discussion). 

Author Response

REVIEWER 1

  1. Throughout the manuscript, the term Empower and its related explanations are repeated several times. I recommend reducing redundancy by avoiding repetitive descriptions and definitions. Consider synthesizing and streamlining mentions of Empower for clarity and conciseness.

-> Response: Thank you for the attention given to our paper. We believe that your comments were very useful in strengthening our paper. We acknowledge that the expression (Empower-Grief) appears substantially throughout the paper. Furthermore, in the introduction, it is also used to describe the original intervention (EMPOWER, as expressed by the original authors). We have revised the text and streamlined the contexts in which it was not essential.

  1. Section 2.3 – Instruments: This section is well-constructed; however, I suggest formatting the content using bullet points or subheadings for each instrument. This would improve readability and allow readers to quickly grasp the structure and content of the measures used.

-> Response: Thank you for your suggestion – we opted for subheadings

  1. Please specify the normality test used to verify the distribution of your data (e.g., Shapiro-Wilk or Kolmogorov-Smirnov). This information is necessary to support the appropriateness of parametric analyses.

-> Response: We thank the reviewer for this observation. However, given that the statistical analyses were conducted entirely within a Bayesian framework, traditional normality tests, such as the Shapiro-Wilk or Kolmogorov-Smirnov tests, were not applied. Bayesian methods, unlike frequentist parametric approaches, do not rely on null hypothesis significance testing or p-values; therefore, they do not require formal testing of normality assumptions prior to analysis. Instead, they model the distribution of the data directly, and robustness to non-normality is a recognised strength of this approach, particularly in small samples.  Please see, for instance, Ghosh, J. K., Delampady, M., & Samanta, T. (2007). An introduction to Bayesian analysis: theory and methods. Springer

  1. Include details of the G*Power analysis performed to justify the use of ANOVA and confirm that the sample size was adequate for detecting effects with sufficient statistical power.

-> Response: We appreciate the reviewer’s comment and ask for understanding regarding our departure from a frequentist approach. A priori power analysis (e.g., via G*Power is a frequentist concept (Cohen, 1988) and is not applicable within the Bayesian analytical framework employed in this study. Bayesian methods, which are now frequently used in psychology research, do not rely on statistical power in the frequentist sense, as they do not involve null hypothesis significance testing or p-values. In fact, many Bayesians explicitly avoid power analysis as it's understood in the frequentist framework. The strength of evidence for or against specific models is evaluated using Bayes Factors, which offer a more nuanced interpretation of results, especially in small-sample, exploratory contexts such as this trial.

Because we understand that when a Bayesian model is evaluated against a frequentist benchmark, readers and reviewers expect a power analysis, we now add a Footnote to the analysis section (Page 7) stating: Power analysis is a frequentist concept. In a Bayesian analysis of actual data, observed power or post hoc power in the frequentist sense is not needed, and it is not used Kruschke, 2014).

Kruschke, J. (2014). Doing Bayesian data analysis: A tutorial with R, JAGS, and Stan.

  1. I recommend checking and clearly reporting the confidence intervals and associated p-values to strengthen the statistical interpretation and ensure transparency in result reporting.

-> Response: Added as suggested (see page 9)

  1. In the discussion section, it would be beneficial to add a dedicated sub-section discussing the implications of the findings and future directions, emphasizing the study’s significance and its potential contributions to both research and clinical practice.

-> Response: Thank you for the sugestion. We have reinforced throughout the discussion the future directions and future research (specific to each aspect of the conclusion). We have also strengthened the argument for clinical practice implications, specifically in the provision of care. With respect to this last idea, and considering your suggestion (below), we have created a dedicated “conclusions section”

  1. In addition, I encourage the authors to include a brief discussion of potentially relevant but not directly analyzed variables, such as personality traits, personal religious beliefs, and the presence of rituals that may assist individuals in processing grief. These factors are often reported in the literature as playing a meaningful role in grief trajectories and coping mechanisms. Here is a list of potential references that may help you (there’s a lot of literature about it, so these references may be a starting point):
    • Gana, K., & K'Delant, P. (2011). The effects of temperament, character, and defense mechanisms on grief severity among the elderly. Journal of affective disorders128(1-2), 128-134. (https://pubmed.ncbi.nlm.nih.gov/20619464/)
    • Mitima-Verloop, H. B., Mooren, T. T., & Boelen, P. A. (2021). Facilitating grief: An exploration of the function of funerals and rituals in relation to grief reactions. Death Studies45(9), 735-745. (https://pubmed.ncbi.nlm.nih.gov/31710282/)

-> Response: These are relevant points. They were added in the Discussion section (third paragraph, p. 11)

  1. It may also be valuable to consider the role of healthcare professionals in providing emotional support. Research suggests that human accompaniment and compassionate care since the diagnosis can significantly support grieving individuals. For instance, even in the devastating context of child loss, many parents do not develop prolonged or complicated grief, especially when surrounded by a supportive and empathetic environment. To further enrich this discussion, you may find it useful to read and cite the following studies and the literature within:
    • Dahò M. (2024) Conscious Decisions in Perinatal Hospice: A Support for Parenting, Family Psychophysical Well-being, and Healthy Grief Processing, Italian Journal of Psychology, 3 (which also includes further references on the role of rituals and personal beliefs). https://www.rivisteweb.it/doi/10.1421/114423
    • Dahò, M. (2020). ‘It was a blanket of love’: How American and Italian parents represent their experience of perinatal hospice through the use of metaphors. Bereavement Care39(3), 112–118. https://www.tandfonline.com/doi/full/10.1080/02682621.2020.1828720 (which explores how grieving parents describe their experiences within a hospice, highlighting the importance of compassionate care by all health workers).

-> Response: The role of the healthcare profession is indeed essential. We have added a comment in the “conclusion” section (p 12). We tried to frame it within the already mentioned conclusions, considering that we do not wish to overreach.

These additions could offer important perspectives that broaden the interpretive depth of the study’s findings. 

  1. Consider adding a dedicated "Limitations" section to discuss methodological or contextual constraints and offer guidance for future research.
  2. Include a Conclusion section that clearly summarizes the main findings, their implications, and potential directions for future investigations. (Last part of the discussion). 

-> Response: Both suggestions were added. See page 12

Reviewer 2 Report

Comments and Suggestions for Authors

Reviewer Comments for Authors

General Comments

This is a well-conducted and clearly written exploratory RCT addressing an important gap in grief interventions: early, structured support for caregivers at moderate risk of Prolonged Grief Disorder (PGD) in palliative care. The manuscript is generally well-organized, with a solid rationale, sound methodology, and a nuanced discussion of findings. However, there are areas where clarity, precision, and additional contextual information could enhance the transparency and impact of the study.

Specific Comments

  1. Introduction

The introduction is generally well-written, providing appropriate and up-to-date references. It clearly outlines the existing literature gap that justifies the study and presents the underlying hypotheses in a coherent and structured manner. However, the reported prevalence rates of Prolonged Grief Disorder (PGD)—notably the 10–20% figure—are not currently supported by a direct bibliographic reference. Please consider adding a citation to substantiate this epidemiological data.

  1. Materials and Methods

While the manuscript provides general information about participant eligibility and inclusion/exclusion criteria, the description of the triage process lacks sufficient detail. Specifically, it is unclear:

  • How participants were screened for eligibility (e.g., was the RAG scale administered via phone, online, or in person?).
  • Who conducted the triage (e.g., trained clinicians, research assistants?).
  • When the triage occurred relative to bereavement and recruitment.
  • Whether any participants were referred by clinicians or if all were recruited exclusively from registries.

A clearer and more structured description of the triage phase would strengthen the methodological transparency and replicability of the study.

Please clarify whether Empower-Grief was delivered in an individual or group format. While the description suggests one-on-one sessions, this is not explicitly stated. Given the potential differences in therapeutic processes and outcomes between individual and group-based interventions—particularly regarding common factors such as peer support—this information is essential for interpretability and replication.

While the Discussion section notes that TAU therapists had substantially more clinical experience than those delivering Empower-Grief, this information should be clearly reported in the Methods section. Therapist expertise and background may influence intervention outcomes and should be described in both arms of the study. Including a summary of therapist training, years of experience, and potential supervision protocols would enhance transparency and allow readers to better assess potential confounding factors.

  1. Results

The results section is clearly structured and generally well reported. The use of Bayesian statistics is a strength of the paper, allowing for a more nuanced interpretation of treatment effects and predictor variables. However, several points require clarification or further emphasis:

The figures currently display group means across time points but do not include indicators of variability (e.g., standard error or confidence intervals). To enhance interpretability—especially given the small sample size and exploratory nature of the study—I recommend updating the figures to include error bars reflecting standard errors or 95% confidence intervals. This would allow for a clearer visual representation of the overlap between groups and the precision of the estimates.

  1. Discussion

The authors propose that similarities in outcomes between Empower-Grief and TAU may be explained by shared common factors such as empathy and emotional support. While this is a plausible interpretation, it would benefit from the inclusion of specific references to support this claim. For instance, Rice (2015) highlights how common therapeutic factors contribute meaningfully to positive outcomes in bereavement groups. Including such references would strengthen the theoretical grounding of this interpretation.

Example reference: Rice, A. (2015). Common therapeutic factors in bereavement groups. Death Studies, 39(1–5), 165–172. https://doi.org/10.1080/07481187.2014.946627.

The study includes a wide range of psychometric instruments targeting theoretically relevant constructs such as psychological flexibility, therapeutic alliance, and coping strategies. However, the discussion does not meaningfully engage with the theoretical implications of these measures or interpret the findings within the framework of the underlying constructs. For instance, the observed role of therapeutic alliance as potential predictor could be contextualized within common factors literature. A deeper theoretical reflection on these dimensions would greatly enrich the interpretation of results and align them with the study’s comprehensive assessment strategy.

Given the exploratory nature of the study and its implementation in a hospital setting, one wonders whether a group-based adaptation of Empower-Grief might have been worth considering. Many of the intervention components—such as psychoeducation, mindfulness, and experiential techniques—are inherently well-suited to group delivery. Group delivery formats are often more efficient in terms of resource allocation, especially in healthcare contexts with limited staffing and time. Additionally, group interventions may leverage therapeutic factors such as shared experience, and peer support, which can be particularly valuable in bereavement work. Even if outcomes remained comparable to TAU, delivering the intervention in a group format might have provided greater feasibility and scalability for routine clinical practice. From a clinical and logistical standpoint, a group format could potentially offer greater efficiency and scalability, even if outcomes remained comparable to TAU. Might this be a direction for future development?

Author Response

REVIEWER 2

  1. Introduction

The introduction is generally well-written, providing appropriate and up-to-date references. It clearly outlines the existing literature gap that justifies the study and presents the underlying hypotheses in a coherent and structured manner. However, the reported prevalence rates of Prolonged Grief Disorder (PGD)—notably the 10–20% figure—are not currently supported by a direct bibliographic reference. Please consider adding a citation to substantiate this epidemiological data.

-> Response: Thank you for pointing this out. Reference added, see page 2.

  1. Materials and Methods

While the manuscript provides general information about participant eligibility and inclusion/exclusion criteria, the description of the triage process lacks sufficient detail. Specifically, it is unclear:

  • How participants were screened for eligibility (e.g., was the RAG scale administered via phone, online, or in person?).
  • Who conducted the triage (e.g., trained clinicians, research assistants?).
  • When the triage occurred relative to bereavement and recruitment.
  • Whether any participants were referred by clinicians or if all were recruited exclusively from registries.

A clearer and more structured description of the triage phase would strengthen the methodological transparency and replicability of the study.

-> Response: This information was added to the “2.4 procedures section” (page 6). The participants were solely retrieved from the registry (as presented in the second sentence of “2.1. Participants”). The moment of contact, relative to the time of death, is framed in the inclusion criteria and then described in the characterisation of the sample. We also clarified the recruitment procedure by adding specific details (see page 6).

Please clarify whether Empower-Grief was delivered in an individual or group format. While the description suggests one-on-one sessions, this is not explicitly stated. Given the potential differences in therapeutic processes and outcomes between individual and group-based interventions—particularly regarding common factors such as peer support—this information is essential for interpretability and replication.

-> Response: Yes. We agree with your argument. This information was explicitly stated in the “2.2 intervention” section (page 4). We now clarify that both interventions were in an individual format.

While the Discussion section notes that TAU therapists had substantially more clinical experience than those delivering Empower-Grief, this information should be clearly reported in the Methods section. Therapist expertise and background may influence intervention outcomes and should be described in both arms of the study. Including a summary of therapist training, years of experience, and potential supervision protocols would enhance transparency and allow readers to better assess potential confounding factors.

-> Response: We have added some information to the interventions to “2.2. Interventions sections”. We do acknowledge that this is an important issue. Please keep in mind that the advantage of the therapists in TAU is against our intervention, and that TAU corresponds to the reality in the hospital in which the study was implemented.

  1. Results

The figures currently display group means across time points but do not include indicators of variability (e.g., standard error or confidence intervals). To enhance interpretability—especially given the small sample size and exploratory nature of the study—I recommend updating the figures to include error bars reflecting standard errors or 95% confidence intervals. This would allow for a clearer visual representation of the overlap between groups and the precision of the estimates.

-> Response: Added as suggested (see page 9)

  1. Discussion

The authors propose that similarities in outcomes between Empower-Grief and TAU may be explained by shared common factors such as empathy and emotional support. While this is a plausible interpretation, it would benefit from the inclusion of specific references to support this claim. For instance, Rice (2015) highlights how common therapeutic factors contribute meaningfully to positive outcomes in bereavement groups. Including such references would strengthen the theoretical grounding of this interpretation.

Example reference: Rice, A. (2015). Common therapeutic factors in bereavement groups. Death Studies, 39(1–5), 165–172. https://doi.org/10.1080/07481187.2014.946627.

-> Response: Considering your comments, we have complemented our explanation, and your suggested reference is a good illustration. We do not aim to make a strong claim considering the limitations of the present study – so we have highlighted the importance of future research. (Discussion second paragraph)

The study includes a wide range of psychometric instruments targeting theoretically relevant constructs such as psychological flexibility, therapeutic alliance, and coping strategies. However, the discussion does not meaningfully engage with the theoretical implications of these measures or interpret the findings within the framework of the underlying constructs. For instance, the observed role of therapeutic alliance as potential predictor could be contextualized within common factors literature. A deeper theoretical reflection on these dimensions would greatly enrich the interpretation of results and align them with the study’s comprehensive assessment strategy.

-> Response: We have expanded our reflection. In any case, we wish to be cautious in what we infer from the results. (Discussion third paragraph – Page 11)

Given the exploratory nature of the study and its implementation in a hospital setting, one wonders whether a group-based adaptation of Empower-Grief might have been worth considering. Many of the intervention components—such as psychoeducation, mindfulness, and experiential techniques—are inherently well-suited to group delivery. Group delivery formats are often more efficient in terms of resource allocation, especially in healthcare contexts with limited staffing and time. Additionally, group interventions may leverage therapeutic factors such as shared experience, and peer support, which can be particularly valuable in bereavement work. Even if outcomes remained comparable to TAU, delivering the intervention in a group format might have provided greater feasibility and scalability for routine clinical practice. From a clinical and logistical standpoint, a group format could potentially offer greater efficiency and scalability, even if outcomes remained comparable to TAU. Might this be a direction for future development?

-> Response: This is a very interesting suggestion. In any case, we still consider the evidence for Empower-Grief to be initial. We are still studying this promising intervention. Indeed, having a group format would be a way to increase effectiveness further. We have included this suggestion (Discussion – fourth paragraph – Page 11)

Reviewer 3 Report

Comments and Suggestions for Authors

This is a modest study addressing a potentially interesting topic. My comments are as follows:

  1. The introduction is too lengthy and should be streamlined to focus on the most relevant background.

  2. Objectives and hypotheses need to be more specific, clearly articulating each dimension assessed by the measures used.

  3. The sample is highly selective, representing only 10% of the database. It currently lacks clarity regarding whether a priori power analysis was conducted or if post-hoc power estimates for the performed analyses are available.

  4. Despite the authors stating no statistically significant differences between the two interventions, they must better support why this particular intervention is promising. What added value does it provide? Given the likely additional costs compared to TAU, without clear efficacy benefits, the rationale for implementation must be justified.

Author Response

REVIEWER 3

This is a modest study addressing a potentially interesting topic. My comments are as follows:

  1. The introduction is too lengthy and should be streamlined to focus on the most relevant background.

-> Response: Thank you for the attention given to our paper. We believe that the revisions following your comments. Regarding the introduction, we have removed sections on the consequences for caregivers and general considerations for prolonged grief disorder (see page 2).

  1. Objectives and hypotheses need to be more specific, clearly articulating each dimension assessed by the measures used.

-> Response:  We have reformulated the goals paragraph at the end of the introduction (see page 4) to balance the need for clarity highlighted by the reviewer with the exploratory nature of this study.

  1. The sample is highly selective, representing only 10% of the database. It currently lacks clarity regarding whether a priori power analysis was conducted or if post-hoc power estimates for the performed analyses are available.

-> Response:  The sample relationship with the population is not uncommon for this type of study, conducted in practice-based settings. The registry often has outdated contacts, and the percentage of eligible participants would necessarily be low. We sought to be transparent about this in the participants section (see page 4) and in the new limitation section (see page 12)

.

With respect to power analysis, Power analysis is a frequentist concept, whereas we are following a Bayesian analytical framework. We understand that when a Bayesian model is evaluated against a frequentist benchmark, readers and reviewers expect a power analysis. But that isn’t applicable. Bayesians explicitly avoid power analysis as it's understood in the frequentist framework. Bayesian methods do not rely on statistical power in the frequentist sense, as they do not involve null hypothesis significance testing or p-values. Instead, the strength of evidence for or against specific models is evaluated using Bayes Factors, which offer a more nuanced interpretation of results, especially in small-sample, exploratory contexts such as this trial.

We now add a Footnote to the analysis section (Page 7) stating: Power analysis is a frequentist concept. In a Bayesian analysis of actual data, observed power or post hoc power in the frequentist sense is not needed, and it is not used Kruschke, 2014).

Kruschke, J. (2014). Doing Bayesian data analysis: A tutorial with R, JAGS, and Stan.

  1. Despite the authors stating no statistically significant differences between the two interventions, they must better support why this particular intervention is promising. What added value does it provide? Given the likely additional costs compared to TAU, without clear efficacy benefits, the rationale for implementation must be justified.

-> Response: We have strengthened the argument  (see page 12). Considering the shorter duration and the ease of involving less experienced therapists, we expect Empower-Grief to have lower costs and therefore be more cost-effective

Reviewer 4 Report

Comments and Suggestions for Authors

This is a professionally conceived and presented study of two interventions for the bereaved. Its goal was not only to compare the 2 programs, but to see whether or how much they actually worked to reduce the incidence of prolonged grief. The study subjects were care-takers of cancer patients, a role known to be long-term stressful. Both programs studied grief soon after death, and with periods of months after. Although the stress of care-taking would have been over for these persons, their unhappiness and grieving was not the focus of hospital care. So 2 psychologically-oriented programs tried to pick up where conventional hospital treatments left off. The researchers found that the differences between the 2 programs were minimal. My sense is that any program directly aimed at surviving ex-caretakers would have helped. And although time by itself does not inevitably heal, to some degree it did. 

I clearly found this study well-done, clear in its goals, and in the end its conclusions made sense. 

Author Response

This is a professionally conceived and presented study of two interventions for the bereaved. Its goal was not only to compare the 2 programs, but to see whether or how much they actually worked to reduce the incidence of prolonged grief. The study subjects were care-takers of cancer patients, a role known to be long-term stressful. Both programs studied grief soon after death, and with periods of months after. Although the stress of care-taking would have been over for these persons, their unhappiness and grieving was not the focus of hospital care. So 2 psychologically-oriented programs tried to pick up where conventional hospital treatments left off. The researchers found that the differences between the 2 programs were minimal. My sense is that any program directly aimed at surviving ex-caretakers would have helped. And although time by itself does not inevitably heal, to some degree it did. 

I clearly found this study well-done, clear in its goals, and in the end its conclusions made sense. 

-> Response: Thank you for your comment. We share your overall appreciation of our data. The reduction of PGD incidence would be an important issue to address in future research. As this was an exploratory study, we focused on reducing symptoms of grief. Thank you for taking the time to review our paper. We have adjusted the discussion to meet your thoughts,

Round 2

Reviewer 1 Report

Comments and Suggestions for Authors

The authors have addressed all my previous suggestions thoroughly. I believe the manuscript is now ready for publication.

Author Response

We sincerely thank the reviewer for their positive feedback and thoughtful contributions throughout the review process. We appreciate your support and are glad the revised manuscript meets your expectations.

Reviewer 3 Report

Comments and Suggestions for Authors

I Thank the authors for the explenation provided in the responde letter, however I don’t feel that the authors have adequately addressed several of my points, specifically the inclusion of study hypotheses and the explanation of why the proposed intervention, although not more effective than TAU, should still be implemented. I find this difficult to understand.
Please expand on these aspects, ideally highlighting the additions in red font so they can be easily identified.

Author Response

We thank the reviewer for their continued attention and comments and we hope to clarify the remaing issues highlighted.

  1. Given the exploratory nature of this trial, our initial intent was to avoid rigid, confirmatory hypotheses. However, we agree that providing clear and transparent expectations is essential for interpretability. Accordingly, we have now reformulated the paragraph at the end of the Introduction section (page 4, highlighted in red font). These reflect our expectations regarding symptom reduction and potential predictors of improvement, while maintaining the exploratory character of the study.
  2. In addition, we have expanded our justification for the potential utility of Empower-Grief in the Discussion and Conclusion sections (pages 11–12, in red font). While the intervention did not show superior clinical outcomes compared to TAU, Empower-Grief offers several practical advantages: it is structured, manualized, brief, and can be delivered by less experienced clinicians with adequate supervision. These features may enhance its scalability and cost-effectiveness, particularly in settings with limited specialised resources.

We hope these revisions adequately address the reviewer’s concerns. All changes are highlighted in red font for ease of identification.